# The Interruption of Transmission of Onchocerciasis in Abia, Anambra, Enugu, and Imo States, Nigeria: The Largest Global Onchocerciasis Stop-Treatment Decision to Date

**DOI:** 10.3390/pathogens13080671

**Published:** 2024-08-08

**Authors:** Cephas Ityonzughul, Adamu Sallau, Emmanuel Miri, Emmanuel Emukah, Barminas Kahansim, Solomon Adelamo, George Chiedo, Samuel Ifeanyichukwu, Jenna E. Coalson, Lindsay Rakers, Emily Griswold, Chukwuemeka Makata, Fatai Oyediran, Stella Osuji, Solomon Offor, Emmanuel Obikwelu, Ifeoma Otiji, Frank O. Richards, Gregory S. Noland

**Affiliations:** 1The Carter Center, Jos 930104, Nigeria; adamu.sallau@cartercenter.org (A.S.); emmanuel.miri@cartercenter.org (E.M.); emmanuel.emukah@cartercenter.org (E.E.); barminas.kahansim@cartercenter.org (B.K.); solomon.adelamo@cartercenter.org (S.A.); george.chiedo@cartercenter.org (G.C.); samuel.ifeanyichukwu@cartercenter.org (S.I.); 2The Carter Center, Atlanta, GA 30307, USA; jenna.coalson@cartercenter.org (J.E.C.); lindsay.rakers@cartercenter.org (L.R.); emily.griswold@cartercenter.org (E.G.); frank.richards@emory.edu (F.O.R.J.); gregory.noland@cartercenter.org (G.S.N.); 3Federal Ministry of Health and Social Welfare, Abuja 900242, Nigeria; ccmakata@gmail.com (C.M.); fatai_oyediran@yahoo.com (F.O.); 4Imo State Ministry of Health, Owerri 460281, Nigeria; stellatosuji40@yahoo.com; 5Abia State Ministry of Health, Umuahia 440236, Nigeria; offorsolomon@yahoo.com; 6Anambra State Ministry of Health, Awka 420110, Nigeria; mail4macho@gmail.com; 7Enugu State Ministry of Health, Enugu 400105, Nigeria; iotiji@yahoo.com

**Keywords:** onchocerciasis, Ov16 ELISA serological assessment, O-150 PCR entomological assessment, interruption of transmission, stop-mass drug administration, elimination

## Abstract

Onchocerciasis causes severe morbidity in sub-Saharan Africa. Abia, Anambra, Enugu, and Imo states of Nigeria were historically classified meso- or hyperendemic and eligible for ivermectin mass drug administration (MDA). After ≥25 years of annual and biannual MDA, serological and entomological assessments were conducted to determine if *Onchocerca volvulus* transmission was interrupted. Dried blood spots collected in October 2020 from ≥3167 children 5–9 years old in each state were screened for *O. volvulus*-specific Ov16 antibody by enzyme-linked immunosorbent assay. Additionally, 52,187 *Simulium damnosum* heads (≥8845 per state) collected over 12 months between 2021 and 2022 were tested by pooled polymerase chain reaction (PCR) for O-150 DNA. Among seven seropositive children, four were found for follow-up skin snip PCR to confirm active infection. Three were negative and the fourth was excluded as he was visiting from an endemic state. The final seroprevalence estimates of each state had 95% upper confidence limits (UCL) < 0.1%. All fly pools were negative by O-150 PCR, giving a 95% UCL infective fly prevalence < 0.05% in each state. Each state therefore met the World Health Organization epidemiological and entomological criteria for stopping MDA effective January 2023. With 18.9 million residents eligible for MDA, this marked the largest global onchocerciasis stop-treatment decision to date.

## 1. Introduction

Onchocerciasis, a parasitic disease caused by *Onchocerca volvulus* (OV) and colloquially known as river blindness (RB), is a neglected tropical disease that causes severe morbidity in sub-Saharan Africa. OV is transmitted through the bites of some species of black flies, the most common vector in Nigeria being *Simulium damnosum* [1]. Adult OV worms can live for up to 15 years in subcutaneous nodules in infected humans, and female worms produce microfilariae (mf) that migrate to the skin. The mf can then be picked up by vectors during blood meals. Mf also trigger host inflammatory reactions that cause severe skin and eye lesions, which can lead to blindness [2].

As of 1995, when the African Program for Onchocerciasis Control (APOC) was launched, an estimated 25% of global OV cases lived in Nigeria [3]. Rapid epidemiological mapping of onchocerciasis (REMO) conducted between 1994 and 1996 indicated that much of Nigeria was endemic with over 50 million people at risk [4]. In 1997, Nigeria—and APOC in general—adopted the community-directed treatment with ivermectin (CDTI) strategy, in which community-directed distributors (CDDs) conduct annual mass drug administration (MDA) with ivermectin (Mectizan^®^, donated by Merck & Co, Inc., Rahway, NJ, USA, known as MSD outside the United States of America and Canada) [5]. Following indications that the elimination of onchocerciasis is feasible using annual and biannual MDA [6,7,8], Nigeria established its National Onchocerciasis Elimination Committee (NOEC) in 2015. In 2017, the NOEC developed the Nigeria Onchocerciasis Elimination Plan (NOEP) [9], a domestication of the 2016 World Health Organization (WHO) onchocerciasis elimination guidelines [10].

Nigeria is a federation comprising 36 states and 1 Federal Capital Territory, subdivided into 774 Local Government Areas (LGAs). For administrative and implementation convenience, the NOEP defines states as the operational onchocerciasis transmission zones. Based on prevalence data from REMO from the mid-1990s and subsequent APOC impact assessments utilizing ONCHOSIM modeling, the NOEP defined the baseline and impact transmission status for the 37 transmission zones and provided guidelines for treatments and stop-MDA assessments [9]. In alignment with WHO 2016 elimination guidelines, impact assessments must meet two criteria before stopping MDA: (1) epidemiological survey among at least 3000 children 5–9 years old must find a Ov16 seroprevalence < 0.1% with 95% confidence; (2) entomological survey must find <0.05% positive black flies with 95% confidence by O-150 enzyme-linked immunosorbent assay (ELISA) polymerase chain reaction (PCR) among at least 6000 flies, or an annual transmission potential (ATP) of <20 with 95% confidence where fewer flies are analyzed [10]. The NOEC also selected epidemiological/serological and entomological assessment sites in each state. The sites include first-line villages (located closest to suspected black fly breeding sites), second-line villages (approximately 15–20 km from first-line villages), and other villages extrapolated from REMO.

### OV Treatment History in Abia, Anambra, Enugu, and Imo States

From 1994 to 1996, the APOC-supported REMO established that much of Abia, Anambra, Enugu, and Imo states had nodular prevalence ≥ 20%, qualifying them as meso- or hyperendemic and meriting inclusion as CDTI priority areas [4]. MDA started in 1995/1996 in meso-/hyper-endemic communities in 72 (88%) of the 82 LGAs in the four states (17 of 17 LGAs in Abia, 16 of 21 LGAs in Anambra, 17 of 17 LGAs in Enugu, and 22 of 27 LGAs in Imo state). MDA was initially delivered by health workers at fixed health facility locations and mobile outreaches until 1997 when, under the CDTI strategy, village volunteers were recruited and trained to conduct house-to-house treatments. The strategy targets 100% geographic coverage of villages and a minimum of 80% program coverage (percentage of eligible population receiving treatment). All persons 5 years old or older are eligible except pregnant women or very sick persons. In Nigeria, 80% of the total population is assumed to be eligible.

Annual MDA expanded to all villages in the 72 endemic LGAs starting in 2015, triggered by the policy shift from control to elimination. There were concerns about loaiasis endemicity in southeast Nigeria in ivermectin-naïve hypoendemic areas, as ivermectin can cause central nervous system adverse events (CNS-AEs) in persons with high-density *Loa loa* microfilaremia [11]. A LoaScope study demonstrated that *Loa loa* microfilaremia in the region was too low to present significant risk of ivermectin-induced CNS-AEs, so MDA expansion to hypoendemic villages was determined to be safe [12].

Lymphatic filariasis (LF) interventions in Nigeria include MDA with ivermectin and albendazole. LF is endemic in 79 (96%) of the 82 LGAs in the four states. Given LF co-endemicity, all communities in all 82 LGAs were receiving annual MDA with ivermectin by 2014.

During 1997 to 2021, all four states reported 100% geographic coverage of MDA in targeted communities and at least 80% program coverage in most years, with a combined cumulative total of 133,655,676 treatments (Figure 1). Program coverage is calculated using census figures plus 2.5% per year projected growth in the population denominator, an imprecise method which contributes to inaccuracy, such as years when the reported program coverage exceeded 100%. Furthermore, coverage reported to the national elimination programme may vary substantially from actual/surveyed coverages [13]. Although the NOEC recommended biannual MDA in 2017, this recommendation was not successfully implemented in all subsequent years due to either drug shortages or the effects of the COVID-19 pandemic. Despite the challenges, ongoing monitoring and mop-up distributions were conducted during MDA to maximize coverage and compliance.

After over 25 years of annual and occasional biannual MDA, in 2020, the NOEC recommended that epidemiological (serological) and entomological assessments be conducted in the four states to determine if transmission was interrupted and MDA could be stopped. This publication reports results of these stop-MDA impact assessments.

## 2. Materials and Methods

The Federal Ministry of Health and Social Welfare (FMOHSW) and Ministries of Health of all four states formally authorized these assessments. As results were not meant to be generalizable, the studies were not considered human research by the Emory Institutional Review Board. Consent (or assent) for the serological and entomological assessments was obtained from community leaders, parents, school children, and adult fly catchers.

### 2.1. OV Stop-MDA Epidemiological Assessments

In October 2020, dried blood spot (DBS) samples were collected from 12,718 children 5 to 9 years old at purposively selected high-risk sites in Abia (3174 children at 27 sites), Anambra (3167 children at 31 sites), Enugu (3180 children at 35 sites), and Imo (3197 children at 40 sites). Sample collection was school based. Primary school enrollment is >80% in all four states [14], and CDDs mobilized unenrolled eligible children to the sample collection site. Short survey forms were completed electronically using open data kit (odk collect) via the NEMO platform (getnemo.org) to gather critical data on the village and the children, such as the GPS location, number of rounds of ivermectin distribution and distance to the nearest streams of the village and the age, sex, ivermectin treatment history, and residency and travel history of the child. Scientists observed applicable standard operating procedures in collecting finger-prick blood on Whatman number 2 filter papers (Sigma-Aldrich, St. Louis, MO, USA).

Samples were transported to The Carter Center laboratory in Jos, Nigeria and stored at −20 °C. The presence of onchocerciasis-specific immunoglobulin G4 antibody was measured using the WHO-approved Onchocerciasis Elimination Program for the Americas (OEPA) version of the Ov16 ELISA [15].

The threshold for stopping MDA is a 95% upper confidence limit (UCL) for prevalence of <0.1% in children aged 5–9 years [9,10]. If there are 10 or more positives by Ov16 ELISA, the transmission zone fails outright; if there are fewer than 10 positives but the 95% UCL does not meet the threshold, follow-up PCR testing of skin snip samples can be conducted on seropositive children to confirm infection status, with PCR-negative children being counted as negative for the final prevalence estimate. We revisited villages in June/July 2021 to trace seropositive children where possible, take skin snip samples, and treat them with ivermectin. After sterile swabbing, two skin snips were taken, one from each iliac crest, using a sterilized size 1.5 mm Holth corneoscleral punch. The samples were preserved in absolute alcohol and analyzed by O-150 PCR at the University of South Florida, USA.

### 2.2. OV Stop-MDA Entomological Assessments

Black flies were collected from July 2021 to June 2022 by human landing capture (HLC) and Esperanza window traps (EWT) from the NOEC-selected sites and additional sites where residents reported a history of black fly bites. Leaders of the communities selected for fly catching gave consent and nominated two locals per site for fly catching. Fly catchers provided informed consent and were treated with ivermectin before beginning. Scientists set up the EWTs and trained the fly catchers on standard techniques for HLC, which include exposing the leg and trapping insects with a glass vial as they land [16]. HLC was carried out for 2 days per week from 7 AM to 6 PM each day at active sites. EWTs used carbon dioxide generated by underground chambers as an attractant and were covered in TangleFoot glue. Attendants collected flies from EWTs twice a week. At each collection the flies are washed in kerosene and preserved in isopropyl alcohol. Health facility staff supervised the fly catchers and collected flies weekly for appropriate storage. Scientists conducted weekly supervision visits to clean traps, service the trap attractant, and send samples to The Carter Center laboratory in Owerri for initial sorting to remove non-vector insects.

Flies were transported monthly to the Carter Center-supported laboratory in Jos, Nigeria, where entomologists sorted *S. damnosum* s.l. flies for O-150 PCR testing. The O-150 PCR procedure and analytical algorithm for calculating prevalence from pooled samples were described previously [17,18]. In brief, *S. damnosum* heads are separated from bodies to detect only the infective L3 stage of the *O. volvulus* parasite in the heads. The heads are pooled by collection site (maximum of 100 per pool) then ground and processed to extract DNA. DNA amplification was conducted by a conventional thermocycler and O-150 was identified by ELISA on the product. Prevalence of infective flies was calculated with both PoolScreen v.2 software and PoolTestR, a package built to translate and expand upon PoolScreen for the R language (R version 4.3.2) [17,19]. The standard algorithm is to calculate the 95% UCL with maximum likelihood estimation when at least one pool is positive and to use Bayesian 95% credible intervals to estimate the 95% UCL when there are no positive pools and frequentist statistical methods are inappropriate. There were no substantive differences in the outputs from the PoolScreen and PoolTestR, with negligible variation in the Bayesian 95% credible intervals only at the hundred-thousandth decimal due to inherent random number generation. The results presented here are the outputs from PoolTestR.

## 3. Results

### 3.1. Epidemiological (Serological) Assessments

A total of 12,718 children were sampled across the four states: Abia (*n* = 3174), Anambra (*n* = 3167), Enugu (*n* = 3180), and Imo (*n* = 3197). A total of seven children were seropositive for Ov16 antibodies: one in Abia (0.03%, 95% UCL: 0.09%), three in Anambra (0.09%, 95% UCL: 0.20%), two in Enugu (0.06%, 95% UCL: 0.15%), and one in Imo (0.03%, 95% UCL: 0.09%). The spatial distribution of sites with positive children is shown in Figure 2; results by LGA are presented in Appendix A.

Abia and Imo states met WHO criteria based on serological results alone (Table 1) [10]. The three seropositive children from Anambra and two from Enugu were sought for skin snip testing. In Anambra, one child was lost to follow-up, one child was PCR negative, and one was PCR positive. The PCR-positive child proved to be a visitor from Ondo, a state where transmission was ongoing as of October 2020; therefore, this child was excluded from Anambra’s calculations. It was assumed conservatively that the child lost to follow up was positive for active infection. The two seropositive children in Enugu were PCR-negative. Therefore, the 95% upper confidence limits of the final seroprevalence estimates for each state were <0.1%: 0.03% (95% UCL: 0.09%) for Abia, 0.03% (95% UCL: 0.09%) for Anambra, 0.00% (95% UCL: 0.06%) for Enugu, and 0.03% (95% UCL: 0.09%) for Imo (Table 1). The NOEC recommended at their May 2021 meeting that all four states proceed to entomological assessments.

### 3.2. Entomological Assessments

There were 213 fly catching sites (51 in Abia, 58 in Anambra, 40 in Enugu, and 64 in Imo), but only 66 (31%) yielded *S. damnosum*: (17/51 [33%] in Abia, 16/58 [28%] in Anambra, 14/40 [35%] in Enugu, and 19/64 [30%] in Imo) (Figure 3 and Appendix A). Productive sites were largely found along a NNW-SSE diagonal associated with the Upper Imo and Upper Orashi rivers and their tributaries. Flies were caught year-round, though there was a notable peak in Anambra in February and in Enugu in March (Figure 4). These patterns diverged from traditional expectations of higher counts during the May to October rainy season [1,20,21].

All flies that were collected and confirmed to be *S. damnosum* s.l. were pool-tested for infective larvae (Table 2). The 52,187 flies analyzed in Abia (*n* = 8845), Anambra (*n* = 11,344), Enugu (*n* = 16,524), and Imo (*n* = 15,474) were all negative for O-150 by PCR, giving 95% UCLs of <1 infective fly per 2000 (i.e., <0.05% infectivity). Details on the results by LGA are included in Appendix A.

## 4. Discussion

Based on NOEP and WHO epidemiological and entomological thresholds [9,10], the results presented here indicate that the transmission of onchocerciasis was interrupted and MDA could therefore be stopped in Abia, Anambra, Enugu, and Imo states of Nigeria. NOEC approved the stop-MDA decision in December 2022 and the FMOHSW announced it during the February 2023 World NTD Day commemoration. Transmission interruption in the four states means that 18.9 million residents were protected from onchocerciasis. This represents the largest one-time stop treatment decision in the history of the global onchocerciasis elimination effort. They join 2.2 million people in Plateau and Nasarawa states where treatments stopped in 2017 and 6.2 million in Kaduna, Kebbi, and Zamfara states where treatments stopped in 2019 [22,23]. This brings the total to 27.3 million treatments stopped in Nigeria, about 49% of the estimated 56 million persons hitherto requiring treatment in the country [24].

Although MDA duration and coverage are not criteria for stop-MDA decisions, inadequate coverage, insufficient rounds of MDA, and emergence of ivermectin resistance could challenge elimination progress. High-coverage annual and occasional biannual MDA were reported in hyper- and meso-endemic communities for over 25 years in each of the four states, well beyond the approximate 15-year life span of the female OV parasite. Hypo-endemic communities only received MDA for about 8 years, but these lower prevalence areas may not have required as many rounds of MDA to interrupt the lower intensity of transmission. Reported coverage could be inaccurate or miss heterogeneity between villages. No research was conducted regarding potential ivermectin resistance in the four states. Given these data limitations, strong post-treatment surveillance (PTS) is a key next step.

There are two primary threats to elimination in the post-treatment phase: recrudescence of transmission due to parasite reservoirs that were not detected during stop-MDA surveys, or importation of parasites in people or black flies moving from endemic areas [25]. Reservoirs may have been missed in Abia, Anambra, Enugu, and Imo due to the sampling strategy. Transmission zones in Nigeria were not delineated based on epidemiological criteria, as transmission was historically too widespread. Instead, they are defined operationally using state administrative boundaries. The average population of each state in Nigeria in 2023 is about 6 million. In sampling only ~3000 children, the studies may have achieved statistical thresholds while missing key clusters of infection in unsampled communities. There may be merit in reducing stop-MDA sampling to sub-state units. The NOEC intentionally identified high-risk villages for sampling, making this a more conservative prevalence estimate than if villages were sampled randomly. However, the risk of missing transmission reservoirs highlights the need for programs to remain active and flexible during PTS. For instance, a report published in late 2022 found OV nodule and mf at meso-/hyperendemic levels in adults in a few sites/villages of Enugu [26]. Our team’s follow-up investigations in and around the same villages demonstrated either no positives or hypo-endemic levels of OV in both adults and children. The NOEC recommended that the villages with mf positives or Ov16-positive children from the follow-up investigations be considered “hot-spots” that should receive twice per year MDA. The issue will be reconsidered when ongoing entomological studies are completed in late 2024.

Abia and Enugu are at greater risk of parasite reimportation than Anambra and Imo given their eastern neighbors, Cross River and Ebonyi states (Figure 3). Cross River and Ebonyi conducted serological assessments twice in the past five years, and on each occasion, failed to meet the stop-MDA threshold. The latest serosurveys, in 2023, had Ov16 ELISA point prevalence of 1.72% and 1.08% among children 5 to 9 years old in Cross River and Ebonyi, respectively, despite more than 20 rounds of ivermectin MDA. The threat from other bordering states is lower. As of December 2023, Benue and Kogi states (to the north of Anambra and Enugu) are classified by the NOEC as having transmission suspected as interrupted, with the most recent epidemiological surveys revealing <1% Ov16 seroprevalence in children. Delta state (to the west of Anambra and Imo states) is classified as transmission interrupted, i.e., it stopped MDA for OV and is itself in PTS. Rivers and Akwa Ibom states (to the south of Abia and Imo states) were classified as non- or hypo-endemic during REMO, though they are due for confirmatory mapping as part of the country’s elimination plan. While international cross-border challenges are well-recognized, large countries with heterogeneity in elimination progress will also face such domestic cross-border challenges. The NOEC established a subcommittee to develop a draft guideline on domestic cross-border activities.

Following a stop-MDA decision, WHO and NOEC guidelines recommend 3 to 5 years of PTS to rapidly detect any recrudescence from transmission reservoirs or reintroduction of parasites [10,24]. Post-treatment activities include public education on stop-MDA, training of CDDs and health facility workers on cross-border monitoring of migrants from states with on-going transmission, and black fly surveillance. Notably, WHO and NOEC guidelines indicate that PTS entomology should not begin until ivermectin MDA for LF has also stopped [9,10]. Abia and Anambra states immediately entered the formal PTS period in January 2023, as all LGAs in the two states previously passed transmission assessment surveys (TAS-1) and stopped LF MDA. Enugu and Imo states have some LF-endemic LGAs in which ivermectin treatments are ongoing as of May 2024. The states cannot formally achieve OV elimination until progress is made by the LF program, highlighting the need to avoid the funding gaps and supply chain limitations that delayed LF TAS.

## 5. Conclusions

Annual and occasional biannual MDA with ivermectin at 100% geographic and 80% program coverage during each round were conducted over a period of about 25 years in Abia, Anambra, Enugu, and Imo states, Nigeria. The data presented here demonstrate that all four states met the WHO serological and entomological criteria for stopping ivermectin MDA for onchocerciasis. Treatment was accordingly halted for 18.9 million persons in the four states. The need for robust PTS activities is highlighted in view of the threat of recrudescence or reintroduction of transmission. The successes reported here demonstrate that Nigeria made great strides toward the goal of OV elimination.

## Figures and Tables

**Figure 1 pathogens-13-00671-f001:**
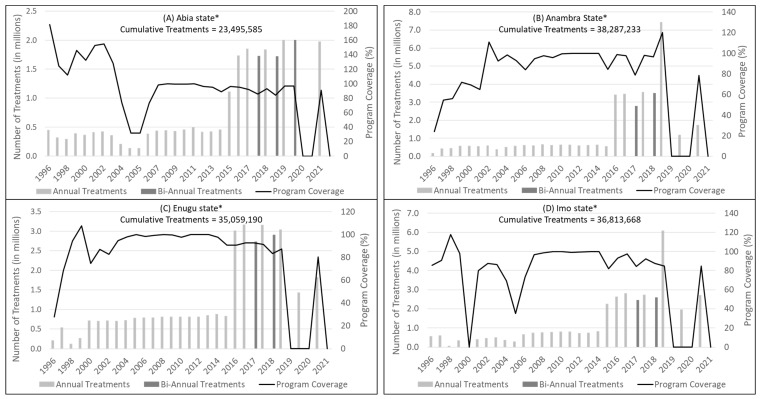
Reported mass drug administration (MDA) treatments and program coverage (1996–2021) in: (**A**) Abia; (**B**) Anambra; (**C**) Enugu; and (**D**) Imo states, Nigeria. * Program approach changed from MDA in hyper- and meso-endemic priority villages to entire district (local government area) populations in 2015.

**Figure 2 pathogens-13-00671-f002:**
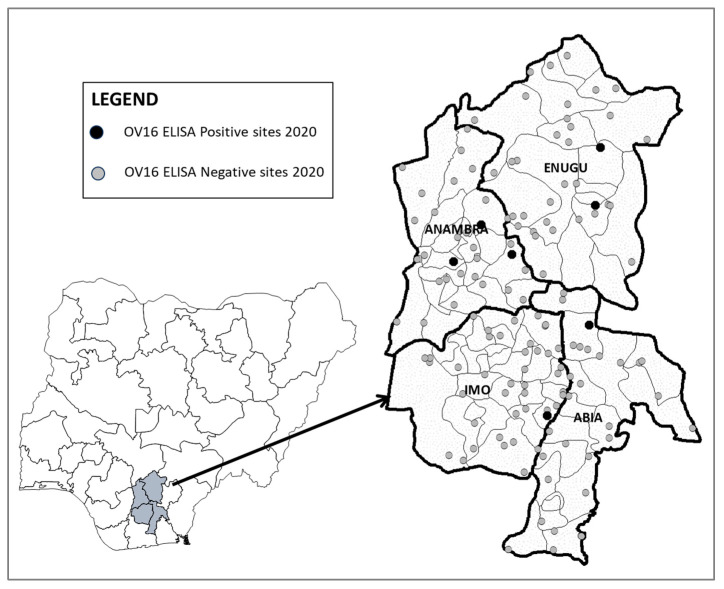
Map of epidemiological assessment sample sites in Abia, Anambra, Enugu, and Imo states, Nigeria, 2020.

**Figure 3 pathogens-13-00671-f003:**
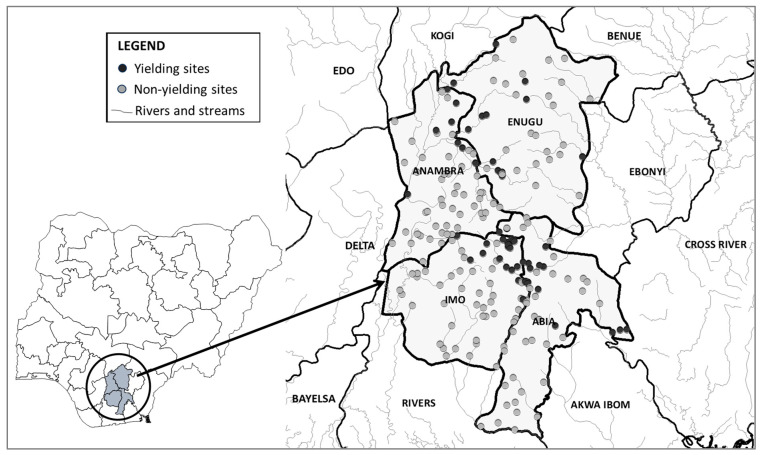
Black fly catching sites in Abia, Anambra, Enugu, and Imo states, Nigeria, 2021–2022.

**Figure 4 pathogens-13-00671-f004:**
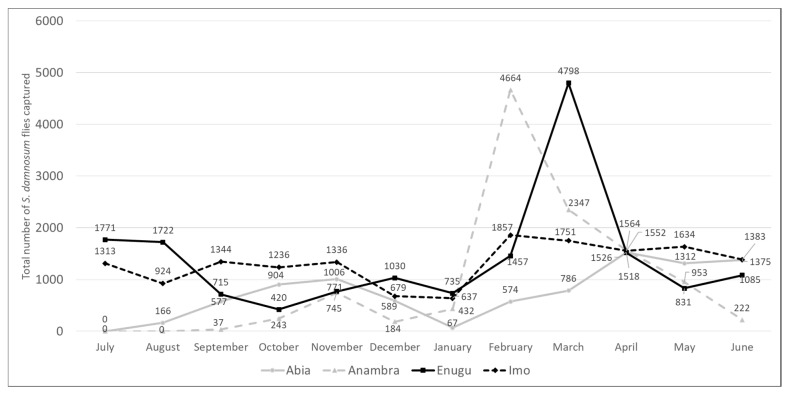
Monthly *Simulium damnosum* s.l. catches in Abia, Anambra, Enugu, and Imo states, Nigeria, 2021–2022.

**Table 1 pathogens-13-00671-t001:** Summary results of onchocerciasis epidemiological assessments in Abia, Anambra, Enugu, and Imo states, Nigeria, 2020–2021.

		Ov16 ELISA Analysis		Follow-Up O-150 PCR Analysis		Final Summary Calculations		
State	Total Tested	Ov16 ELISA Positive (*n*)	Ov16 ELISA Positive % (95% UCL)	Children with Skin Snips Tested by O-150 PCR (*n*)	O-150 PCR Positive (*n*)	Count of OV-Positive Children (*n*)	Final Proportion Positive % (95% UCL)	**Conclusion**
Abia	3174	1	0.03% (0.09%)	0	n/a	1	0.03% (0.09%)	Pass, proceed to entomological assessment
Anambra	3167	3 *	0.09% (0.20%)	2	1 *	1 *	0.03% (0.09%)	Pass, proceed to entomological assessment
Enugu	3180	2	0.06% (0.15%)	2	0	0	0.0% (0.06%) **	Pass, proceed to entomological assessment
Imo	3197	1	0.03% (0.09%)	0	n/a	1	0.03% (0.09%)	Pass, proceed to entomological assessment

Abbreviations: ELISA = Enzyme-linked immunosorbent assay; n/a = not applicable; OV = *Onchocerca volvulus*; PCR = polymerase chain reaction; and UCL = upper confidence limit. * One of the three children with positive Ov16 ELISA results from Anambra could not be found during follow-up for skin snip collection and subsequent PCR testing. The only child with a PCR-positive skin snip proved to be visiting from an endemic focus in Ondo state and was therefore excluded from the Anambra sample. The child that could not be found was conservatively assumed to be positive in the final calculations. Thus, the final proportion positive and 95% UCL values were calculated as 1 positive from a total sample of 3166. ** Estimated using Bayesian credible interval due to all samples being negative.

**Table 2 pathogens-13-00671-t002:** Summary results of onchocerciasis entomological assessments in Abia, Anambra, Enugu, and Imo states, Nigeria, 2021–2022.

State	Sites Yielding Black Flies (n/N)	Total Black Flies (*n*)	Pools of 100 Flies Tested (*n*)	Pools of <100 Flies Tested (*n*)	Pools Positive by O-150 PCR (*n*)	Estimated Prevalence of OV Infective Larvae (*n*, 95% UCL) *	Conclusion **
Abia	17/51	8845	84	17	0	0 per 2000 (UCL: 0.022)	Pass
Anambra	16/58	11,344	106	16	0	0 per 2000 (UCL: 0.017)	Pass
Enugu	14/40	16,524	159	21	0	0 per 2000 (UCL: 0.016)	Pass
Imo	19/64	15,474	143	28	0	0 per 2000 (UCL: 0.012)	Pass

Abbreviations: MDA = mass drug administration; NOEC = [Nigeria] National Onchocerciasis Elimination Committee; OV = *Onchocerca volvulus*; PCR = polymerase chain reaction; UCL = upper confidence limit; and WHO = World Health Organization. * Prevalence estimates from pooled testing scheme are calculated using PoolTestR and were confirmed against results from PoolScreen software. See Methods for details. ** The WHO and NOEC criterion for passing Stop-MDA onchocerciasis entomology assessments is <1 infective black fly per 2000 (i.e., <0.05%) with 95% confidence.

## Data Availability

Restrictions apply to the datasets used in this study. Data supporting this study are owned by the Nigeria National Onchocerciasis Elimination Program. Requests to access the datasets should be directed to the corresponding author, who can initiate contact with the Nigerian Federal Ministry of Health and Social Welfare to set up appropriate data sharing agreements for access.

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
