# Peer review of "The Interruption of Transmission of Onchocerciasis in Abia, Anambra, Enugu, and Imo States, Nigeria: The Largest Global Onchocerciasis Stop-Treatment Decision to Date"

_pathogens, 2024, doi:10.3390/pathogens13080671_

Round 1

Reviewer 1 Report

Comments and Suggestions for Authors

Ityonzughul and colleagues presented data about Onchocerca volvulus infection in humans and vectors after 25 years of MDA in 4 Nigerian states. The results showing that the transmission of O. volvulus is interrupted that allows the stop of the MDA under the WHO epidemiological and entomological criteria. The presented results are important for the research community and the manuscript is well written. I only have minor issues about the manuscript .

1) Do the authors have any information about the compliance of MDA in the 4 states. In other countries like Cameroon MDA is also applied for more than 15 years but transmission was not interrupted probably also due to compliance issues and drug resistance (are there any information about resistance in the 4 states?). I think these two points need to be discussed if possible to show how these obstacles were successfully tackled in the 4 states.

2) The supplement data includes comments and Table S2 is empty. 

Reviewer 2 Report

Comments and Suggestions for Authors

Overall the conclusions of this manuscript appear valid.  I think the major issue with the manuscript as written is that it appears to be designed to use the maximum amount of words possible. I suspect if it were edited for readability and to reduce redundancy the entire manuscript could easily be 25% or even 50% shorter and more concise. I cant tell if AI (Chatbot) was used to try to make the text longer by suggesting repetitive word choices.  Regardless of this please make it concise and unrepetitive.

Readability and flow are choppy and sometimes hard to follow.  I give some examples here.

Line 26-27:  You say that “ O. volvulus-specific Ov16 antibody was detected” in over 12,000 people by ELISA.  In reading the paper I think you mean that over 12,000 people were screened for OV16 antibodies by ELISA.  Otherwise you are indicating a massive seroprevelance.

Line 43 and elsewhere: Avoid using redundant noun series or using genera/species as adjectives. “ Simulium black flies” for example.  Well 100% of all Simulium are black flies.  So why the redundant wording. 

Line 44-45: The sentence following the sentence about the flies implies without further clarity that these nodules are in the flies.  Clarity here is best.

Line 164: “Qualified laboratory scientists” remove the “Qualified laboratory”

The redundant wording is either due to a desire to extend the length of the manuscript or is a product of AI (Chat GPT) helping with the writing.  I found most paragraphs and sometimes every sentence just wordy beyond the need. Please edit throughout. The examples above are just a few of many.

I would suggest that the whole manuscript be carefully red for clarity and flow of information and logic.

Is there a journal policy about the IRB?  I ask because Line 143-149 might be unnecessary if the journal already requires an IRB for human studies and it would be implied. The same goes for line 156.  This seems redundant and per my initial statement this manuscript is very redundant in the text. The redundancy if edited could greatly reduce the length and thus reduce cost, save unneeded pages, and make tis overall easier to read. If these statements are all needed then keep them but per the Line 143-149 you already said the approvals were received.

Line 164-167: Same you state that they followed the applicable SOPs and then spell them out.

Line 194” “blackfly” change to “black fly”

Comments on the Quality of English Language

I don't think the problems are with the science but the paper itself is a slog to go through because it repeats itself or just uses 5 words when 2 would suffice.
